# Prevalence and Significance of Autoantibody Seropositivity in Children with Wilson’s Disease

**DOI:** 10.3390/diagnostics13040768

**Published:** 2023-02-17

**Authors:** Wojciech Jańczyk, Joanna B. Bierła, Ilona Trojanowska, Aldona Wierzbicka-Rucińska, Bożena Cukrowska, Piotr Socha

**Affiliations:** 1Department of Gastroenterology, Hepatology, Nutritional Disorders and Pediatrics, Children’s Memorial Health Institute, Al. Dzieci Polskich 20, 04-730 Warsaw, Poland; 2Department of Pathomorphology, Children’s Memorial Health Institute, Al. Dzieci Polskich 20, 04-730 Warsaw, Poland; 3Department of Biochemistry, Radioimmunology and Experimental Medicine, Children’s Memorial Health Institute, Al. Dzieci Polskich 20, 04-730 Warsaw, Poland

**Keywords:** autoantibody, anti-nuclear antibody, anti-smooth muscle antibody, anti-tissue transglutaminase 2 antibody, autoimmune hepatitis, transient elastography

## Abstract

Autoantibodies occur in healthy subjects as well as in children with Wilson’s disease (WD), but their prevalence and significance are unknown. Thus, we aimed to assess the prevalence of autoantibodies and autoimmune markers, and their relationship to liver injury in WD children. The study included 74 WD and 75 healthy children as a control group. Patients with WD underwent transient elastography (TE) examinations, as well as determination of liver function tests, copper metabolism markers, and serum immunoglobulins (Ig). In the sera of the WD patients and controls, anti-nuclear (ANA), anti-smooth muscle, anti-mitochondrial, anti-parietal cell, anti-liver/kidney microsomal, anti-neutrophil cytoplasmic autoantibodies, and specific celiac antibodies were determined. Among the autoantibodies, only the prevalence of ANA in children with WD was higher than in the controls. There was no significant relationship between the presence of autoantibodies and liver steatosis or stiffness after TE. However, advanced liver stiffness (E > 8.2 kPa) was related to IgA, IgG, and gamma globulin production. The type of treatment did not influence the prevalence of autoantibodies. Our results suggest that autoimmune disturbances in WD might not be directly related to liver damage as expressed by steatosis and/or liver stiffness after TE.

## 1. Introduction

Wilson’s disease (WD) is a rare, autosomal recessive genetic disorder of copper metabolism leading to progressive and toxic accumulation of copper mainly in the liver and nervous system. It is caused by mutations in the ATP7B gene encoding a copper-transporting ATPase, which binds copper with ceruloplasmin in the hepatocyte and enables copper excretion into the bile [1,2,3,4,5].

The predominant manifestation of disease in children is liver injury, which ranges from asymptomatic, through steatosis and hepatitis to advanced fibrosis, cirrhosis, and acute liver failure. Liver dysfunction may occur as early as 1 year of age, but is rarely symptomatic before the age of 5 [3,6,7]. A diagnosis of WD in children should be based on the Ferenci scoring system, which includes, e.g., clinical symptoms, serum ceruloplasmin, 24-h urinary copper, copper in the liver tissue, and mutation analysis of the *ATPB7* gene [3,8]. 

WD is a progressive hepatopathy, so monitoring of the liver function is mandatory during treatment, optimally with non-invasive techniques. Liver biopsy (LB) may be recommended only at diagnosis, but not routinely, in order to assess liver damage. Especially in children, the use of LB is limited as it carries a risk of pain, bleeding, and sedation [9].

Transient elastography (TE, FibroScan, Echosens, France) is a non-invasive tool to quantitively assess liver stiffness/fibrosis (LSM) and steatosis (CAP, controlled attenuation parameter) [10]. TE has been widely validated in adults with chronic liver diseases [10,11,12,13,14,15]. There are also an increasing number of studies in children that confirm the good diagnostic performance of TE for predicting the degree of liver fibrosis and steatosis [16,17,18].

There are reports that WD may coexist with or mimic autoimmune disorders such as autoimmune hepatitis [19,20]. In selected cases, differential diagnosis may be challenging [21,22]. 

Autoimmune antibodies occur commonly in autoimmune hepatitis (AIH), but may be present in other liver diseases, as well as in healthy subjects [23,24,25,26,27]. In some conditions, they may be associated with a more severe course of disease [28,29]. Low-titer autoantibodies have been found in patients with WD [20], but their prevalence and significance regarding liver function and the presence of steatosis or fibrosis is unknown. Autoimmune markers have not been extensively and systematically investigated in WD so far.

Consequently, in our study, we aimed to assess the prevalence of autoantibodies, autoimmune markers, and their relationship to liver injury in consecutively enrolled children with WD.

## 2. Materials and Methods

### 2.1. Patients

We included 74 children (41 girls) with a mean age of 11.6 years with WD. The diagnosis was based on the scoring system, which included parameters of copper (Cu) metabolism (serum level of ceruloplasmin and Cu level in daily (24 h) urine collection), and/or molecular examination for *ATP7B* gene mutations [8]. Fulminant or acute liver failure (increased INR > 1.5) was the exclusion criterion. Patients included into this trial regularly attended our outpatient clinic for control visits between January 2018 and January 2020. They were treated with either zinc or d-penicillamine. 

The control group included 75 healthy children with a mean age of 12.6 years (42 girls) recruited from the relatives of the employees.

### 2.2. Transient Elastography

All WD patients underwent transient elastography by FibroScan^®^ 502 Touch (Echosens, Paris, France) using 3.5-MHz medium (M) probe. The examinations were performed in fasted patients, in the right liver lobe. Liver stiffness (LSM, kPa) was obtained using the velocity of a low-frequency (50 Hz) elastic shear wave propagation through the liver [10]. Liver steatosis by CAP (dB/m) was measured by liver ultrasonic attenuation at 3.5MHz [30]. The final LSM And CAP results were presented as the median of 10 valid measurements. They were regarded as reliable when the success rate exceeded 60% and the interquartile range/median (IQR/M) ratio was below 30%. We considered cut-off values of CAP >250 dB/m as suggestive for liver steatosis. Values of LSM >6 kPa were regarded for liver fibrosis and >8.2 kPa were regarded for advanced fibrosis [17,31].

### 2.3. Detection of Autoantibodies

The following autoantibodies of class immunoglobulin G (IgG) were analyzed in the sera of WD patients and controls with the use of an indirect immunofluorescence technique: antinuclear antibodies (ANA), anti-smooth muscle antibodies (SMA), anti-mitochondrial antibodies (AMA), anti-parietal cell antibodies (APCA), anti-liver/kidney microsomal antibodies type 1 (LKM-1), and anti-neutrophil cytoplasmic antibodies (ANCA). According to ESPGHAN recommendations [23], the presence of ANA was detected using commercial testing slides with Hep-2 cell line (EuroImmun, Lubeck, Germany) at 1:20 screening dilution. Positive results for the ANA tests were confirmed by the immunoblot method (ANA3b test, EuroImmun, Germany). SMA, AMA, APCA, and LKM-1 were evaluated on a commercially-available rat tissue substrate (liver, kidney and stomach; BioSystems, Barcelona, Spain) at a 1:20 screening dilution. 

For ANCA detection, commercial slides with Hep-2 cell and neutrophils fix with formalin and ethanol were used (EuroImmun, Germany) at a 1:10 screening dilution. 

All of the specimens positive at 1:10 or 1:20 dilution were retested with the use of two-fold dilutions to determine the final antibody titer. Microscopic assessment was done by two independent diagnosticians. 

For detection of celiac-specific antibodies, the presence of anti-tissue transglutaminase 2 (anty-tTG2 IgA) autoantibodies in class IgA and anti-deamidated gliadin peptide (DPG) antibodies in class IgG were analyzed with the use of an automated Thermo Scientific Phadia 100 system (Phadia, Sweden). According to the manufacturer’s protocols, an antibody concentration >10 and <7 U/mL was considered as positive and negative, respectively. Positive results were confirmed by testing anti-endomysial (EuroImmun, Germany) antibodies (EMA) in both IgA and IgG classes.

### 2.4. Clinical Measurements

Laboratory tests, including liver function tests (alanine transaminase (ALT), aspartate transaminase (AST)), cholesterol (CHOL), triglycerides (TG), immunoglobulins, and serum proteins, were performed at the same time. Copper metabolism markers (ceruloplasmin, 24 h-urinary copper excretion) at the time of diagnosis were analyzed.

### 2.5. Statistical Analysis 

We presented parameter characteristics as medians with an interquartile range, because we did not find a normal distribution in most of the parameters studied. For data comparison between groups, the Mann–Whitney U test was applied. We grouped categorical data in tables and performed either chi-square or two tailed Fischer exact test for comparison. We regarded p values <0.05 as statistically significant. Statistics were performed with Statistica version 13 (TIBCO software Inc., Palo Alto, CA, USA).

### 2.6. Bioethics Approval

The study was approved by the local Bioethical Committee N/DBW/591/W/2017 and written informed consent was obtained from the patients’ parents, caretakers, or patients aged ≥ 16 years old, concerning the use of their serum samples for scientific purposes. The study protocol conformed to the ethical guidelines of the 1975 Declaration of Helsinki.

## 3. Results

### 3.1. Patients’ Characteristics

Patients with WD presented with slightly elevated liver enzymes. In most of them, ceruloplasmin and 24-h copper excretion were abnormal at the time of diagnosis.

Fibroscan^®^ examination showed a normal median liver stiffness of 4.6 kPa and slightly increased liver steatosis CAP-245dB/m. Eight patients (11%) had advanced fibrosis (LSM > 8.2 kPa).

The main class immunoglobulins and gamma globulins remained within the normal range. Kayser–Fleischer rings were seen in 10% of patients. Patients were treated predominantly with zinc. Detailed characteristics of the patients are presented in Table 1.

### 3.2. Autoantibodies in WD and Controls

The presence of autoantibodies in the WD group was demonstrated in 62 patients (84%), of which 27 patients (37%) showed a highly positive reaction (titer 1:160 and higher). In contrast, only 21 out of 74 (28%) control patients showed the presence of autoantibodies. The difference between the WD group and controls was statistically significant (*p* < 0.00001). In the control group, high positivity of the reaction was not shown. Fifteen patients with WD (20%) were both ANA and SMA positive.

#### 3.2.1. Antinuclear Antibody, ANA 

We found a significantly higher frequency of positive ANA antibodies (titer (1:20); *n* = 53 vs. *n* = 20, *p* < 0.0001) in WD patients compared with the controls (Table 2). 

When the patterns of ANA were analyzed, the speckled ANA was seen almost two and a half times more often in WD children than in the control group (*n* = 49 vs. *n* = 20, *p* < 0.00001). Among WD patients with speckled ANA, two patients presented with a spindle fibers pattern (NuMa-like) and one with nucleolar patterns. Twenty-one patients (28%) were highly ANA positive (=>1:160), predominantly females (67%). In contrast, homogenous ANA, although detected only in WD patients (*n* = 4), did not achieved statistical significance in comparison with the controls (*p* = 0.12). Among these patients, we found one patient with a “rings and rolls” pattern. Only two patients were highly ANA positive.

In the control immunoblot, only one patient with a homogenous ANA pattern showed an equivocal value of scan intensity 6 (equivocal range of test was 5.1–10) for the dsDNA antibodies. 

Moreover, we found four WD patients who showed a strong non-specific nuclear glow, while in the control immunoblot, we received only one patient positive result for nucleosomes (value 22 of scan intensity, when the positive limit value was >10).

#### 3.2.2. Tissue Autoantibodies and Anti-Neutrophil Cytoplasmic Antibodies (ANCA)

Although the WD children more frequently showed the presence of SMA and APCA antibodies than the control group, the differences in positivity were not statistically significant (Table 3). Among the WD children, we found one serum sample with anti-glomerular basement membrane antibodies and in two serum samples we detected a signal that corresponded to the bile ducts (Figure 1).

We did not detect the presence of AMA or LKM-1 antibodies in any serum sample.

We detected one WD patient (girl; 16.3 years old) who was pANCA positive in 1:10 dilution with a final positive titer of 1:320. In the control group, we did not find any ANCA positive reaction.

#### 3.2.3. Celiac Specific Antibodies

We detected one patient (13.6 year old girl) who tested positive for tTG2 IgA (60 U/mL) and DPG (283 U/mL). Further diagnostics also revealed positive EMA. Eventually, she was diagnosed with celiac disease. Among the other WD patients there was another child with previously confirmed celiac disease already on gluten-free diet.

### 3.3. Relationship of Autoantibodies to Laboratory Tests

In the studied group, there were no statistically significant associations between ANA or SMA positivity and the results of the liver function tests, lipids, or copper metabolisms markers (ceruloplasmin and 24 h urinary copper excretion). The presence of ANA was markedly associated only with IgA levels (*p* = 0.02) (Table 4).

### 3.4. Transient Elastography vs. Liver Function Tests and Autoimmune Markers

Liver stiffness was not correlated with liver function tests, copper metabolism markers, or main class immunoglobulins. Notably, the presence of specific autoantibodies (e.g., ANA) was not related to liver steatosis or stiffness after transient elastography (Fibroscan^®^).

However, patients with liver fibrosis after TE had significantly higher AST and ALT levels (*p* < 0.01). Advanced liver stiffness (E > 8.2 kPa) was related to ALT (*p* = 0.02), IgA, IgG, and gamma globulins production (*p* < 0.01). Statistical analysis showed that liver stiffness was not correlated with copper metabolism markers or the main class immunoglobulins. 

Liver steatosis (CAP) was significantly related to ALT activity (*p* = 0.03). The presence of steatosis was not related to the presence of autoantibodies of any specific type.

### 3.5. Impact of Treatment on Autoimmune Markers

In our group, there were 23 patients treated with d-penicillamine, 48 with zinc, and 3 were naive. 

Both groups did not statistically differ regarding liver function tests or liver steatosis after TE. Interestingly, children who took d-penicillamine had higher liver stiffness (*p* = 0.05) than the others.

In addition, the type of treatment did not influence the prevalence of autoantibodies. ANA were positive in the majority of patients treated with d-penicillamine (82.6%), but were also commonly prevalent in zinc-treated children (70.8%).

## 4. Discussion

For the first time in the literature, we assessed the prevalence of autoantibodies in a large cohort of patients with WD. We found that the majority (84%) of patients were positive for ANA, ASMA, APCA, ANCA, or a combination of autoantibodies that are also implicated in AIH or other autoimmune diseases. Of these, 27 patients (37%) showed a highly positive reaction (titer 1:160 and higher). In contrast, only 28% of healthy control children were positive for any autoantibodies. 

ANA occurred with a higher prevalence in WD than in the healthy controls in both slightly (1:20) (70.6% vs. 26.6%) and highly positive (≥1:160) ANA patients (27.9% vs. 2.7%) (*p* < 0.05). These results suggest that positive autoantibodies are far more common in children and adolescents with WD than in the healthy population [32]. Furthermore, their prevalence seems to be even higher than in other liver diseases such as non-alcoholic fatty liver disease (NAFLD) or viral hepatitis [29,33,34,35,36]. Surprisingly, the percentage of autoantibody seropositivity in our patients was approaching that seen in autoimmune diseases [23,37]. However, we did not report elevated gammaglobulins and IgG, nor AIH type 2-specific antibodies similar to anti-LKM. 

On the other hand, the prevalence of autoantibodies in our healthy control group remained in agreement with previous reports. As observed earlier, the percentage of the population with ANA was approximately up to 25% when using indirect immunofluorescence microscopy performed on HEp-2 cells [32,33,34], whereas in our cohort, 26.6% was observed. According to some reports, using IIFA (indirect immunofluorescence assays), ANA in low counts may appear in up to 40% of healthy people [38]. The incidence of a significantly elevated ANA level (≥1:160) in the general population is 2.5% [32], whereas it was 2.7% in our healthy controls. Some data suggest that the prevalence of ANA increases with age [39].

There is also an increasing number of studies in children that confirm the good diagnostic performance of TE for predicting the degree of liver fibrosis and steatosis [16,17,18]. TE has been widely validated in adults with chronic liver diseases [10,11,12,13,14,15]. We found that specific autoantibody positivity (e.g., ANA and SMA) was not more frequent in patients with/without liver fibrosis or steatosis after TE. Therefore, we did not confirm that the presence of autoantibodies may be associated with a more severe course of disease in our patients, as it has been shown in some other chronic liver disorders such as HCV or NAFLD [28,29]. Autoantibody positivity was not related to liver function tests, copper metabolism markers, or other metabolic/immunological disturbances, with the exception of the ANA to IgA level (Table 4).

Autoimmune disorders have become more common recently and may affect up to 5% of the population. Autoimmune antibodies are produced subsequent to the humoral immune response, against self-cellular proteins and nucleic acids [40]. They are well recognized as serological hallmarks of autoimmune diseases. Autoantibodies are essential in diagnosing autoimmune liver diseases, including AIH and primary biliary cholangitis (PBC). In particular, the “homogenous” ANA pattern is the most specific of AIH, especially in combination with actin type SMA [41,42]. On the contrary, PBC is characterized by a rim-like/membranous immunofluorescence pattern and the “multiple nuclear dots” ANA pattern [43,44].

Autoantibody positivity in the context of liver diseases without AIH has been a topic of scientific interest for years. The prevalence of autoantibody positivity has been thoroughly investigated in NAFLD. According to existing evidence, it ranges from 21% to 36% among adults with NAFLD [29,45,46,47]. However, the relationship between autoantibody positivity and severity of liver disease seems inconsistent. Few studies in adults have revealed an association between autoantibody positivity and fibrosis severity, necroinflammation, and ANA positivity [29,48] Adams et al. found that positive autoantibodies were associated with a more severe course of NAFLD expressed by higher fibrosis stage, higher inflammatory grade, and hypergammaglobulinemia [29]. In contrast, other studies have shown that patients with NAFLD and positive autoantibodies were less likely to have an advanced steatosis [47]. In one of a few pediatric reports, Patton et al. failed to show the association between autoantibody positivity and histologic patterns of liver steatosis or fibrosis [49]. In a recent paper, Yodoshi et al. reported the prevalence of 33% ANA and SMA in children with NAFLD, with a positive association between ANA and severe steatosis [34].

ANA and SMA may be present in several other acute and chronic liver diseases such as viral hepatitis B, C (HCV), D, and E (HEV); acute liver failure; alcohol-induced liver disease; hepatocellular carcinoma; and drug-induced liver injury (DILI) [24,50,51,52]. In HCV infection, ANA, SMA, and LKM antibodies can be detected with a prevalence of about 30%. Patients with ANA more frequently presented with advanced fibrosis and necroinflammation in the liver than those without ANA [35,36]. Among the patients with acute HEV infection, Terziroli et al. found that half of them had least one autoantibody and 33% had ANA. In the follow-up (median 12 months), nobody from the autoantibody positive patients developed AIH [53]. Certain drugs causing DILI are commonly associated with the formation of serum autoantibodies [54]. In the Drug-Induced Liver Injury (DILI) Network prospective study, de Boer et al. found positive ANA and SMA (in 72% and 60%, respectively) at the onset of DILI caused by nitrofurantoin, minocycline, methyldopa, or hydralazine [55].

In WD, the presence and significance of autoimmune factors has not been extensively investigated so far. Czlonkowska et al. demonstrated immunological abnormalities in WD caused by liver cirrhosis and disturbance in copper metabolism. They found increased levels of IgG and IgM, as well as a statistically lower percentage of T-cells in WD and liver cirrhosis than in the control group [56].

Low-titer autoantibodies have been found in patients with WD without a diagnosis of AIH, but their prevalence and significance is unknown [20,57]. A wide array of autoantibodies may be present in WD coexisting with autoimmune disorders such as systemic lupus erythematosus (SLE). In these cases, the LKM-1 antibody, SMA, anti-actin antibodies, ANA, antiphospholipid antibodies, and even anti-double-stranded DNA were detected [58,59,60]. Sometimes, the presence of these autoantibodies may lead to a false diagnosis of autoimmune disease. Up to date, a few cases of superimposed AIH and WD have been reported [19,20,21,22,61].

In our opinion, the high prevalence of autoantibodies seen in our patients with WD did not indicate an autoimmune disease. We consider this finding a phenomenon that should rather be treated as a benign condition. Although most patients were positive for ANA or SMA, none of them were finally diagnosed with autoimmune disease or AIH, which remains in line with previous observations in adults [62]. Interestingly, two patients were confirmed to have celiac disease. Therefore, based on our results, it is hard to prove a higher prevalence of autoimmune disorders in WD. Some authors have suggested that autoantibodies in WD may be potentially induced by hepatocyte necrosis, especially in the early stages of this disease [61]. As we did not perform liver biopsy at the time of the study, we cannot confirm this hypothesis. The association between ANA positivity and liver fibrosis in our cohort also remains to be further evaluated, as most of our patients presented with mild liver injury.

However, there is also ongoing debate about whether autoantibodies may be observed in the serum of patient long before the diagnosis of autoimmune disease. In SLE and Sjögren’s syndrome, ANA is present even for many years before the onset of the first symptoms of the disease [63,64]. On the contrary, many studies have shown that ANA-positivity in the general population does not predict an autoimmune disease in the future [65,66]. In our patients with high titers of autoantibodies, further monitoring for the development of autoimmune disorders could be advised.

Furthermore, although WD is a disorder of a well-established pathogenesis, there have been some unresolved issues regarding its natural history [3,4,5]. The clinical presentation is largely variable, ranging from presymptomatic, to liver disease and/or to the central nervous system. It is unclear why WD may become symptomatic at any age and present with hepatic or neurologic disease. The common presentation in children is liver inflammation and steatosis, whereas neurologic symptoms are rarely seen. Still, there may be some genetic and environmental factors that influence the course of disease [67,68]. In this context, an interesting concept has emerged suggesting that immune factors might be responsible for triggering inflammatory process in various conditions, including liver disorders [69,70].

Although we could not demonstrate a significant relationship between ANA/SMA and transaminases, all patients who were ANA positive (≥1:160) presented with elevated ALT. Furthermore, we found a significant correlation between ANA and IgA levels. Recently, Stremmel et al. published a case report describing a 9-year-old girl with presymptomatic WD with persistently elevated transaminases, positive for ANA without hyperglobulinemia [71]. They also analyzed a large group of similar patients with various liver disorders who had elevated ALT, normal IgG, but positive ANA. They accounted for a total of 20% of the ANA positive cohort, which was significantly higher than the AIH group (*p* < 0.00001). The authors subsequently suggested a hypothesis of an immune inbalance could trigger liver disease, but a more detailed analysis of this entity is required.

In WD, autoimmune-related adverse events such as SLE, dermatomyositis, or arthritis may potentially result from chelator therapy, especially d-penicillamine [8,23,60,62,72,73]. Seesle et al. reported that 2.6% of their WD cohort developed an autoimmune disease after long-term d-penicillamine treatment [62]. Past reports have suggested that ANA titers increased during d-penicillamine therapy [74]; however, this statement was questioned later [62]. Up until now, these questions have not been addressed in the pediatric population. In our cohort, 23 patients were treated with d-penicillamine. We found that 82% of them were ANA positive, but a similar prevalence was observed in patients treated with zinc (72%). Therefore, it seems that d-penicillamine did not induce an autoimmune response to a greater extent than zinc in our patients. Furthermore, we demonstrated that the type of therapeutic agent (d-penicillamine or zinc) was not related to the presence of specific autoantibody or abnormalities in liver function tests. We took into account that our pediatric cohort was probably not treated for long enough to develop d-penicillamine immune-related adverse-events. However, we found that liver stiffness after TE was significantly increased in the d-penicillamine-treated children. This could have resulted from the selection of patients, as in our center, d-penicillamine is usually administered in patients with advanced liver disease.

Additionally, we also assessed the prevalence of celiac-specific autoantibodies in patients with WD, which again has never been investigated. Thanks to this screening, we found a patient with positive anti-tissue transglutaminase antibodies in whom celiac disease was confirmed. A second patient with a previous diagnosis of celiac disease was already on gluten-free diet. This accounted for, in total, 2.7%. Because of the small numbers used, we could not make any conclusion about the associations of WD with celiac disease, but this would be interesting to investigate further.

Finally, our study stressed the importance of a careful differential diagnosis of WD, particularly with AIH. There have been several case reports of the coexistence of both diseases [3,19,20,21,61]. In general, WD can be distinguished from AIH, but in some circumstances, a diagnosis is challenging [22,75]. This is especially demanding and critical in the stage of acute liver failure (ALF). The presence of autoantibodies such as ANA and SMA, as well as abnormal liver function tests and liver fibrosis, may occur in both diseases, as we showed in our cohort. Screening for autoantibodies in unknown hepatitis is mandatory; however, they occur in various liver disorders other than AIH and in healthy individuals, and are usually not diagnostic on their own. In clinical practice, the diagnosis of WD may be overlooked in patients with a provisional diagnosis of AIH, ineffectively treated with steroids and/or azathioprine [22]. Therefore, it is important to remember that a diagnosis of AIH still requires the exclusion of WD.

The strengths of this study include the large, well described cohort of children with WD, the cross-sectional design, and comparison with a control group of healthy patients. The prevalence and significance of autoantibodies have never been investigated in pediatric WD.

Some limitations of our study can be related to selection of patients with mild liver disease and the absence of histological examination. Because a liver biopsy was not conducted in our patients, we used TE. It is a non-invasive procedure to quantitatively assess liver stiffness (fibrosis) and steatosis, and seems ideally tailored to the pediatric population. TE has already been applied and validated in patients with various chronic liver diseases [10,11,12,13,14,15,16,17,76,77,78]. The cut-off points for degree of fibrosis and steatosis vary depending on the underlying liver disease. As there are no reference values of TE in WD, we arbitrarily chose the well-established cut-off points for liver stiffness and steatosis in chronic liver diseases [17,31].

## 5. Conclusions

According to our findings, the prevalence of tissue autoantibodies in children with WD seems to be higher than in the general population. Our results suggest that autoimmune disturbances in WD might not be directly related to liver damage as expressed by steatosis and/or liver stiffness examined using TE. We also showed that patients with Wilson’s disease may have a higher prevalence of celiac specific autoantibodies than in the general population. We suggest that patients with suspected or confirmed AIH should always undergo detailed diagnostics towards WD.

## Figures and Tables

**Figure 1 diagnostics-13-00768-f001:**
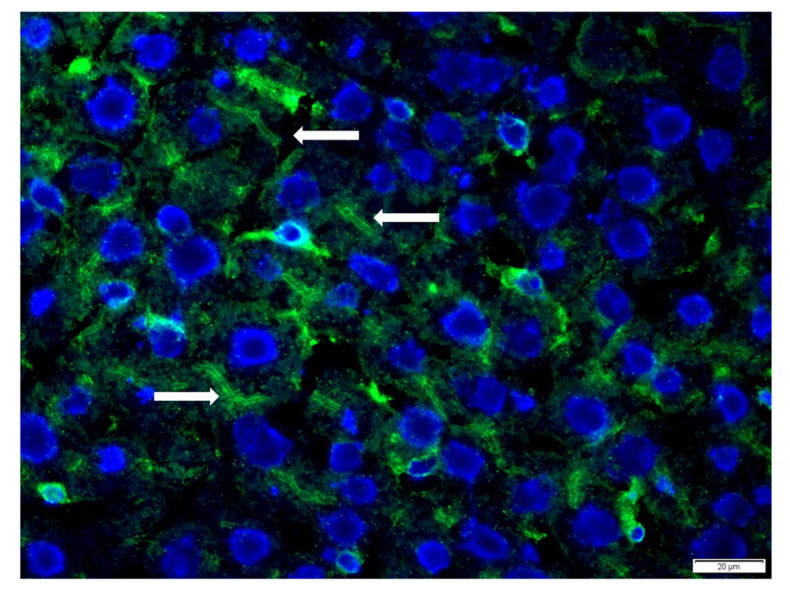
Representative microscopic view of the immunofluorescence reaction Wilsons’s disease patient antibodies against bile ducts (white arrow). The reaction was done on commercially-available rat tissue substrate (liver, BioSystems, Barcelona, Spain). For visualization, secondary antibodies against human IgG conjugated with FITC and nuclear DAPI staining were used.

**Table 1 diagnostics-13-00768-t001:** Characteristics of patients with Wilson’s disease.

Parameter	*n* = 74
Age (years)	11.6
Gender, *n* (%)	males 33 (44) females 41 (55)
ALT (U/L)	49 (11–303)
AST (U/L)	34 (13–220)
CHOL (mg%)	164 (107–252)
TG (mg%)	112 (37–491)
IgA	1.4 (0.2–4.1)
IgG	9.9 (4.9–16.5)
IgM	0.9 (0.3–2.4)
Ceruloplasmin (mg%)	0.14 (0.02–0.27)
24-h copper excretion (µg/24 h)	88 (1–979)
Autoantibodies positive, *n* (%) ANA SMA APCA	62 (84) 54 (73) 29 (39) 6 (8)
LSM (kPa)	4.6 (2.6–15.3)
CAP (dB/m)	245 (105–369)
Patients with liver steatosis, *n* (%)	34 (46)
Patients with fibrosis, *n* (%)	13 (18)
Patients with advanced fibrosis, *n* (%)	8 (11)
Treatment, n (%) - zinc - d-penicilamine - naive	48 (64) 23 (31) 3 (4)

All data are presented as median (range), unless stated otherwise. ALT, alanine transaminase; AST, aspartate transaminase; CHOL, cholesterol; TG, triglycerides; LSM, liver stiffness/fibrosis; CAP, liver steatosis (controlled attenuation parameter).

**Table 2 diagnostics-13-00768-t002:** Immunofluorescent antinuclear antibody (ANA) pattern found in the control/healthy and Wilson’s disease children.

ANA Pattern	Titer	1:20		1:40		1:80		1:160		1:320	
		WD Group	Control Group	WD Group	Control Group	WD Group	Control Group	WD Group	Control Group	WD Group	Control Group
Homogenous	positive	4 (.3%)	0	4 (5.3%)	0	4 (5.3%)	0	3 (4%)	0	2 (2.7%)	0
	slightly positive	0	0	0	0	0	0	1 (1.3%)	0	1 (1.3%)	0
Speckled	positive	49 (65.3%)	20 ≥ (26.6%)	27 (36%)	11 (14.7%)	16 (21.3%)	2 (2.7%)	10 (13.3%)	0	5 (6.7%)	0
	slightly positive	0	0	22 (29.3%)	9 (12%)	2 (2.7%)	9 (12%)	7 (9.3%)	2 (2.7%)	5 (6.7%)	0

Statistical analysis was performed with use of the two-tailed Fisher’s exact test *p* < 0.01. Statistical differences were found for speckled ANA in titer ≥1:20.

**Table 3 diagnostics-13-00768-t003:** Immunofluorescent antibody pattern found in control/healthy and Wilson’s disease children.

Pattern	Titer	1:20		1:40		1:80	
		WD Group	Control Group	WD Group	Control Group	WD Group	Control Group
SMA	positive	21 (28%)	15 (20%)	6 (8%)	4 (5.3%)	4 (5.3%)	0
	slightly positive	0	0	15 (20%)	11 (14.6%)	1 (1.3%)	3 (4%)
APCA	positive	6 (8%)	6 (8%)	3 (4%)	1 (1.3%)	3 (4%)	0
	slightly positive	0	0	3 (4%)	5 (6.7%)	0	0

Statistical analysis was performed with use of the two-tailed Fisher’s exact test. There was no statistical significance between the studied groups. SMA, anti-smooth muscle antibodies; APCA, anti-parietal cell antibodies.

**Table 4 diagnostics-13-00768-t004:** Comparison of the clinical and laboratory findings between patients with Wilson’s disease grouped by anti-nuclear antibody (ANA) status.

Variable	ANA Positive (*n* = 54)	ANA Negative (*n* = 20)	*p* Value
Age (y)	11.9	11.2	>0.05
Females, *n* (%)	31 (57)	10 (50)	>0.05
**Lab tests**			
ALT (U/L)	51 (26,89)	47 (29,71)	0.98
AST (U/L)	34 (24,52)	36 (28,40)	0.60
CHOL (mg%)	164 (144,181)	168 (154,187)	0.47
TG (mg%)	113 (90,150)	110 (90,150)	0.83
IgA	1.3 (0.9,1.6)	1.7 (1.2,2.4)	**0.02**
IgG	9.6 (8.3,11.2)	10.4 (9.4,12.1)	0.21
IgM	1 (0.7,1.4)	0.8 (0.6,1)	0.07
Gammaglobulins	9.9 (8.4,12)	11.4 (8.6,13.7)	0.42
Ceruloplasmin	0.1 (0.1,0.18)	0.11 (0.09,0.16)	0.22
24 h urine copper excretion	87 (62,140)	94 (64,112)	0.43
**Transient elastography**			
LSM (kPa)	4.4 (3.6,5.3)	5.2 (4.3,6.4)	0.07
CAP (dB/m)	241 (209,264)	259 (234,282)	0.22

Data are presented as *n* (%) for categorical variables and medians and interquartile ranges for non-categorical variables (for statistical significance *p* < 0.05). ALT, alanine transaminase; AST, aspartate transaminase; CHOL, cholesterol; TG, triglycerides; LSM, liver stiffness/fibrosis; CAP, liver steatosis (controlled attenuation parameter).

## Data Availability

The data presented in this study are available on request from the corresponding author.

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
