# Peer review of "Prevalence and Significance of Autoantibody Seropositivity in Children with Wilson’s Disease"

_diagnostics, 2023, doi:10.3390/diagnostics13040768_

Round 1

Reviewer 1 Report

Respected authors This study you have done about Wilson's disease, which is rare in the society, starting from childhood and can lead to serious complications when the diagnosis and treatment is delayed, which we see in adulthood, is very well planned, done and written very well. What's even better is that no close relationship could be established between autoimmune markers and Wilson's disease, except for ANA, and this article is very well stated. This study has completed an important deficiency in this regard. I believe that this study will be beneficial for both medical students, residents and specialists. I wish you success in your work.

Author Response

Respected authors This study you have done about Wilson's disease, which is rare in the society, starting from childhood and can lead to serious complications when the diagnosis and treatment is delayed, which we see in adulthood, is very well planned, done and written very well. What's even better is that no close relationship could be established between autoimmune markers and Wilson's disease, except for ANA, and this article is very well stated. This study has completed an important deficiency in this regard. I believe that this study will be beneficial for both medical students, residents and specialists. I wish you success in your work.

Answer:

We would like to thank you very much for such a positive review. We are much obliged to you for your work and we feel greatly appreciated.

Reviewer 2 Report

A very interesting study and it is in line with the overall prevalance of autoantibodies in general population

It would be interesting to study the same in patients of ACLF or ALF

Author Response

A very interesting study and it is in line with the overall prevalance of autoantibodies in general population

It would be interesting to study the same in patients of ACLF or ALF

Answer:

We would like to thank you very much for such a positive review.

Reviewer 3 Report

In this study, the authors aimed to assess the prevalence of autoantibodies and autoimmune markers and their relationship to liver injury in Wilson’s disease (WD) children. They studied 74 WD and 75 healthy children, as a control group. Patients with WD underwent transient elastography (TE) examinations, and determination of liver function tests, copper metabolism markers, serum immunoglobulins (Ig). In sera of WD patients and controls anti-nuclear (ANA), anti-smooth muscle , anti-mitochondrial, anti-parietal cell, anti-liver/kidney microsomal, anti-neutrophil cytoplasmic autoantibodies, and specific celiac antibodies were determined.

Only ANA were more prevalent in children with WD than in controls. There was no significant relationship between the presence of autoantibodies and liver steatosis or stiffness on TE. However, advanced liver stiffness (E>8.2kPa) was related to IgA, IgG and gammaglobulins production. Type of treatment did not influence the prevalence of autoantibodies. They concluded that autoimmune disturbances in WD might not be directly related to liver damage as expressed by steatosis and/or liver stiffness on TE. 

The study is of interest, however, the authors should further discuss the diagnostic significance of their findings according to literature data. In particular, the authors should further discuss the diagnostic relevance of autoantibodies in liver disease recalling the well-defined diagnostic role of non-organ-specific autoantibodies in autoimmune hepatitis (AIH) and primary biliary cholangitis (PBC). In particular, not all ANA and SMA have equal diagnostic accuracy and significance. In particular, it is well-known that the "homogenous" immunofluorescence ANA pattern is the most specific of AIH, as well as SMA exhibiting the "anti-actin" pattern, as reported in previous studies (Diagnosis and therapy of autoimmune hepatitis. Mini Rev Med Chem. 2009 Jun;9(7):847-60; Antibodies to filamentous actin (F-actin) in type 1 autoimmune hepatitis. J Clin Pathol. 2006 Mar;59(3):280-4).

On the contrary, in PBC, it is well-known that the so-called ANA PBC-specific are those characterized by the rim-like/membranous immunofluorescence pattern and the "multiple nuclear dots" ANA pattern, as previously demonstrated (Antinuclear antibodies as ancillary markers in primary biliary cirrhosis. Expert Rev Mol Diagn. 2012 Jan;12(1):65-74; Antinuclear antibodies giving the 'multiple nuclear dots' or the 'rim-like/membranous' patterns: diagnostic accuracy for primary biliary cirrhosis. Aliment Pharmacol Ther. 2006 Dec;24(11-12):1575-83.).

This may help in a proper interpretation of autoantibody positivity in patients with liver diseases.

Author Response

The study is of interest, however, the authors should further discuss the diagnostic significance of their findings according to literature data. In particular, the authors should further discuss the diagnostic relevance of autoantibodies in liver disease recalling the well-defined diagnostic role of non-organ-specific autoantibodies in autoimmune hepatitis (AIH) and primary biliary cholangitis (PBC). In particular, not all ANA and SMA have equal diagnostic accuracy and significance. In particular, it is well-known that the "homogenous" immunofluorescence ANA pattern is the most specific of AIH, as well as SMA exhibiting the "anti-actin" pattern, as reported in previous studies (Diagnosis and therapy of autoimmune hepatitis. Mini Rev Med Chem. 2009 Jun;9(7):847-60; Antibodies to filamentous actin (F-actin) in type 1 autoimmune hepatitis. J Clin Pathol. 2006 Mar;59(3):280-4).

On the contrary, in PBC, it is well-known that the so-called ANA PBC-specific are those characterized by the rim-like/membranous immunofluorescence pattern and the "multiple nuclear dots" ANA pattern, as previously demonstrated (Antinuclear antibodies as ancillary markers in primary biliary cirrhosis. Expert Rev Mol Diagn. 2012 Jan;12(1):65-74; Antinuclear antibodies giving the 'multiple nuclear dots' or the 'rim-like/membranous' patterns: diagnostic accuracy for primary biliary cirrhosis. Aliment Pharmacol Ther. 2006 Dec;24(11-12):1575-83.).

This may help in a proper interpretation of autoantibody positivity in patients with liver diseases.

Answer:

We would like to thank you very much for all your valuable comments regarding our article.

In verse 282 it was:

“They are well recognized as serological hallmarks of autoimmune diseases. Autoantibodies are essential in diagnosing autoimmune liver diseases, including AIH and primary biliary cholangitis.”

Although we investigated children and PBC was not considered in differential diagnosis, we agree that pointing out diagnostic role of autoantibodies in AIH and PBC will add valuable context to our discussion. Therefore we corrected the text with suggested literature (verses 282-286):

“They are well recognized as serological hallmarks of autoimmune diseases. Autoantibodies are essential in diagnosing autoimmune liver diseases, including AIH and primary biliary cholangitis (PBC). Particularly "homogenous" ANA pattern is the most specific of AIH, especially in combination with actin type SMA [41,42]. On contrary, PBC is characterized by the rim-like/membranous immunofluorescence pattern and the "multiple nuclear dots" ANA pattern [43,44]”

  1. Granito, A.; Muratori, P.; Ferri, S.; Pappas, G.; Quarneti, C.; Lenzi, M.; Bianchi, F.B.; Muratori, L. Diagnosis and therapy of autoimmune hepatitis. Mini Rev. Med. Chem. 2009, 9(7),847-860.
  2. Granito, A.; Muratori, L.; Muratori, P.; Pappas, G.; Guidi, M.; Cassani, F.; Volta, U.; Ferri, A.; Lenzi, M.; Bianchi, F.B. Antibodies to filamentous actin (F‐actin) in type 1 autoimmune hepatitis. J. Clin. Pathol. 2006, 59(3), 280–284.
  3. Granito, A.; Muratori, P.; Muratori, L.; Pappas, G.; Cassani, F.; Worthington, J.; Guidi, M.; Ferri, S.; Molo, C.DE.; Lenzi, M.; Chapman, R.W.; Bianchi F.B. Antinuclear antibodies giving the 'multiple nuclear dots' or the 'rim-like/membranous' patterns: diagnostic accuracy for primary biliary cirrhosis. Aliment Pharmacol Ther. 2006, 24(11-12),1575-1583
  4. Granito, A.; Muratori, P.; Quarneti, C.; Pappas, G.; Cicola, R.; Murato, L. Antinuclear antibodies as ancillary markers in primary biliary cirrhosis. Expert Rev. Mol. Diagn. 2012, 12(1), 65-74.